# Diversity and New Species of Ascomycota from Bamboo in China

**DOI:** 10.3390/jof10070454

**Published:** 2024-06-28

**Authors:** Zhiying Zhao, Taichang Mu, Nemat O. Keyhani, Huili Pu, Yongsheng Lin, Ziying Lv, Jinming Xiong, Xiaohao Chen, Xinyang Zhan, Huajun Lv, Motunrayo Yemisi Jibola-Shittu, Peisong Jia, Jianlong Wu, Shuaishuai Huang, Junzhi Qiu, Xiayu Guan

**Affiliations:** 1State Key Laboratory of Ecological Pest Control for Fujian and Taiwan Crops, College of Life Sciences, Fujian Agriculture and Forestry University, Fuzhou 350002, China; zhaozhiyingv1230@163.com (Z.Z.); mutaichang@163.com (T.M.); hdpuhuili@163.com (H.P.); linyongsheng0909@163.com (Y.L.); 18359311723@163.com (Z.L.); serendipity879440255843996@outlook.com (X.C.); 1468550268@139.com (X.Z.); 13328626122@163.com (H.L.); motunjibolashittu@gmail.com (M.Y.J.-S.); 2Department of Biological Sciences, University of Illinois, Chicago, IL 60607, USA; keyhani@uic.edu; 3Institute of Microbiology, Chinese Academy of Sciences, Beijing 100101, China; xiongxiaojing1016@gmail.com; 4Institute of Plant Protection, Xinjiang Academy of Agricultural Sciences, Urumqi 830091, China; jps-fly@163.com; 5Landscape Management Department, Xiamen Botanical Garden, Xiamen 361004, China; sega999@163.com; 6School of Ecology and Environment, Tibet University, Lasa 850000, China; alaxender1989@126.com; 7Key Laboratory of Ministry of Education for Genetics, Breeding and Multiple Utilization of Crops, College of Horticulture, Fujian Agriculture and Forestry University, Fuzhou 350002, China

**Keywords:** *Apiospora*, *Bifusisporella*, bambusicolous fungi, molecular phylogeny, morphology, new species

## Abstract

Bamboo is an economically important crop that has gained prominence as an alternative to wood to reduce deforestation and ecosystem destruction. Diseases of bamboo that typically occur on leaves and stems can cause significant loss, reducing the quality and yield of the bamboo. However, there are few reports identifying the fungal species diversity and potential pathogens of bamboo. Here, we describe four new species of plant fungi from the leaves of bamboo within Fujian provinces, China. Fungi were isolated from diseased leaves collected within Fujian province and identified based on their morphological characteristics and multilocus phylogenies using nucleotide sequences derived from combined datasets of the intervening 5.8S nrRNA gene (ITS), the 28S large subunit of nuclear ribosomal RNA gene (LSU), the large subunit of RNA polymerase I (*rpb1*), the translation elongation factor 1-α gene (*tef1-α*), and the partial beta-tubulin gene (*tub2*). These analyses helped reveal and clarify taxonomic relationships in the family Magnaporthaceae. The new species of bambusicolous fungi identified include two species of *Bifusisporella*, described as *B. fujianensis* sp. nov. and *B. bambooensis* sp. nov., and two species of *Apiospora*, described as *A. fujianensis* sp. nov. and *A. fuzhouensis* sp. nov. This study further expands the characterization and distribution of fungi associated with bamboo.

## 1. Introduction

The bamboo plant (Poales, Bambusoideae) family includes over 1400 different species of monocotyledon, mostly evergreen perennials. Bamboo encompasses the largest members of the grass family, occur naturally in a wide range of different ecosystems, and are cultivated as a highly versatile crop [1]. Bamboos are well known to confer a number of beneficial ecological effects including carbon sequestration and erosion control, include some of the fastest growing plants known, have widespread ornamental use, and represent an important economic crop in regions where they are cultivated. Commercial applications of bamboo as a material for use in building/construction and fabrication of furniture, fabric, paper, cookware, cooking utensils, and many other items stems from its high strength-to-weight ratio and ease of cultivation that includes rapid plant growth. In addition, the plant is a food source for humans and other animals, notably giant and red pandas, as well as bamboo lemurs [2]. Currently, approximately 80% of the world’s bamboo species are found in the eastern and/or southern areas of Asia, with China having the richest bamboo resources in terms of highest diversity and overall cultivated area, accounting for more than 50% of the worldwide bamboo species [3,4]. Bamboos plants show high resistance to microbial diseases, with fungal ascomycetes as the major microorganisms that limit the health and productivity of bamboo forests. Different bamboo fungi can infect various parts of the plant, resulting in nevus (gall-like “tumors”), spotted wilt, leaf damage/necrosis, and other symptoms and diseases that can lead to reduced quality and yield of the bamboo. Deleterious effects of Bambusicolous Ascomycota can impact economic development, and methods for the biological control of some bamboo fungi can reduce losses in bamboo forests and the cultivation industry, helping to maintain diversity, plant populations, and the varied beneficial ecological functions of bamboo forests [5]. Over 1150 different species of ascomycetes may have some association (including pathogens, mutualists, and commensals) with bamboo, of which 350 asexual morphs, 240 hyphomycelia, and 110 coelomycetes have been tentatively identified [6]. These fungi are mainly distributed in the Sordariomycetes, Dothideomycetes, and Eurotiomycetes, with the more representative families of bamboo fungi found in the Magnaporthaceae and Apiosporaceae families within the Sordariomycetes.

Magnaporthaceae classification was proposed by Cannon [7] and included the genus *Magnaporthe* and its related genera *Buergenerula*, *Clasterosphaeria*, *Gaeumannomyces*, *Herbampulla*, and *Omnidemptus*. More recently, additional taxa classified within Magnaporthaceae include *Magnaporthiopsis* [8], *Bussabanomyces*, *Kohlmeyeriopsis* and *Slopeiomyces* [9], *Pseudophhialophora* [10], *Falciphora* [11], *Neogaeumanyces* [12], *Budhanggurabania* [13], *Falciphoriella* and G*aemannomycella* [14], and *Bifusisporella*. Currently, Magnaporthaceae consists of 25 genera and more than 100 species. The genus *Bifusisporella* was erected by Rejane [15], with *B. sorghi* designated as the type species. The morphology of *Bifusisporella* is characterized by septate, branched mycelium with a smooth, hyaline to light brown surface, conidiophores reduced to conidiogenous cells, which can be solitary or aggregated, curved and elongated cylindrical or clavate, and are typically light brown. Conidia are described as dimorphic, with macroconidia slightly more curved than microconidia, and both sickle-shaped, hyaline, and smooth [15].

The Apiosporaceae fungal family belongs to the Ascomycota (Sordariomycetes Amphisphaeriales), with the type genus being *Apiospora* Sacc. introduced by Saccardo [16] and the type species being *A. montagnei* Sacc. The sexual morphology of *Apiospora* is characterized by hyaline ascospores surrounded by thick gelatinous sheaths [17,18,19]. The asexual form of *Apiospora* is characterized by lenticular conidia that are spherical or subglobose and usually light brown to brown in color [20,21]. Most *Apiospora* species are associated with plants as endophytes, with some species being economically important plant pathogens [22,23].

In this study, four fungal species, two of which represent new species, found growing on bamboo plants were identified and placed within the Magnaporthaceae, with their taxonomic placement determined based on morphological characteristics and molecular identification. The latter involved multilocus phylogenetic reconstructions using a combined dataset of the intervening 5.8S nrRNA gene (ITS), the 28S large subunit of nuclear ribosomal RNA gene (LSU), the large subunit of RNA polymerase I (*rpb1*), and the translation elongation factor 1-α gene (*tef1-α*) nucleotide sequences. Similarly, four additional fungal isolates (again, with two representing new species) were identified and placed within the Apiosporaceae based upon morphological characteristics and molecular taxonomic and phylogenetic analyses using the combined marker loci sequence dataset (ITS + LSU + *tef1-α* + *tub2*). Our results identify two new species of *Bifusisporella*, *Bifusisporella fujianensis* sp. nov. and *Bifusisporella bambooensis* sp. nov. (Magnaporthaceae), and two new species of *Apiospora*, *Apiospora fujianensis* sp. nov., *Apiospora fuzhouensis* sp. nov. (Apiosporaceae), which are illustrated and described. This study expands the diversity of fungi infecting the economically and environmentally important bamboo plant.

## 2. Materials and Methods

### 2.1. Fungal Isolates and Morphology

Specimens were collected from diseased bamboo leaves in groves located in Fujian province, China. Tissue fragments with a total area of about 25 mm^2^ were removed from the edges of the bamboo leaves in which disease spots, e.g., necrosis, wilting, and/or discoloration/blackening, were apparent. Samples were soaked in 75% ethanol for 45–60 s, then soaked in sterile deionized water for 45 s and washed with sterile water. Tissue fragments were then transferred to a 5% sodium hypochlorite solution for 30 s, followed by three washes in sterile deionized water for 60 s. The fragments were dried with sterilized filter paper and then transferred to the PDA plates which were incubated at 25 °C for 5–7 days following previously established procedures [24]. Growing edges of fungal mycelia were transferred to new PDA plates and plates were incubated for 5–7 d. The procedure was continued until the fungal culture was pure (typically 2–4 times). To promote sporulation and observe the colony morphology, purified isolates were inoculated in the center of PDA and synthetic low-nutrient agar (SNA) plates and cultured at 25 °C under alternating conditions of 12 h near-ultraviolet light and 12 h dark [25]. At 7 and 14 d of growth on PDA, photos of the colonies were taken with a digital camera, and the morphology of conidiomata, conidiophores, and conidiogenous cells was observed using a stereomicroscope (Nikon SMZ74, Tokyo, Japan). Samples were also prepared for analyses by scanning electron microscope (SEM, Nikon Ni-U; HITACHI SU3500) as described [26]. Fungal micromorphology and structure were measured by Digimizer 5.4.4 software. Single colony purified cultures were cut and stored in 10% sterilized glycerin and sterile water at 4 °C for future detailed study.

### 2.2. DNA Extraction, PCR Amplification, and Sequencing

Genomic DNA was isolated from fresh mycelia using a fungal DNA extraction mini ki, from cells cultured at 25 °C on PDA for 15–30 days as described [27]. Primers ITS5/ITS4 [28], LROR/LR5 [29], RPB1-Ac/RPB1-Cr [30,31], EF1-983F/EF1-2218R [32], EF1-728F/EF-2 [33], and Bt2a/Bt2b [34] were used for amplification of the intervening 5.8S nrRNA gene (ITS), the 28S large subunit of nuclear ribosomal RNA gene (LSU), the large subunit of RNA polymerase I (*rpb1*), the translation elongation factor 1-α gene (*tef1-α*),and the partial beta-tubulin gene (*tub2*) by polymerase chain reactions (PCR) as described [27]. Primer sequences are given in Table 1.

PCR amplification of target loci was performed using a Bio-Rad thermal cycler (Hercules, CA, USA) with a 25 μL reaction volume of 12.5 μL 2×Rapid Taq Master Mix (Vazyme, Nanjing, China), with 1 μL (10 μM) for the forward and reverse primers (Sangon, Shanghai, China) and 1 μL for the template genomic DNA in the amplifier, and adjusted with distilled deionized water to a total volume of 25 μL. PCR products were visualized on 1% agarose gel electrophoresis. Bidirectional (both strand) sequencing of PCR products was conducted by the Tsingke Company Limited (Fuzhou, China). Consensus sequences were assembled using MEGA 7.0 [35]. New sequences generated in this study were uploaded to GenBank (https://www.ncbi.nlm.nih.gov, accessed on 19 March 2024, Table 2).

### 2.3. Phylogenetic Analyses

Based on maximum likelihood (ML) and Bayesian inference (BI) analyses, phylogenetic trees were constructed to explore the phylogeny relationships of the fungal strains, grouping them into either the Magnaporthaceae or Apiosporaceae families. Corresponding gene loci of the reference sequences were downloaded from GenBank. *Ophioceras dolichostomum* (CBS 114926) was selected as an outgroup taxonomic unit for the phylogeny of Magnaporthaceae, and *Sporocadus trimorphus* (CBS 114203) was selected as an outgroup taxonomic unit for the phylogeny of Apiosporaceae. All sequences were aligned using the MAFFT v. 7 online service (http://mafft.cbrc.jp/alignment/server/, accessed on 2 February 2024) [36] and manually adjusted in BioEdit v.7.2.6.1 [37] and MEGA 7.0 [35].

In addition, four simultaneous Markov Chain Monte Carlo (MCMC) chains, starting with 2,000,000 generations of random trees, were sampled every 100th generation, resulting in a total of 20,000 trees. The first 25% of trees were discarded as burn-in of each analysis. Branches with significant Bayesian Posterior Probabilities (BYPP > 0.90) were estimated in the remaining 15,000 trees [38]. Phylogenetic trees were plotted with FigTree v.1.4.4 [39] and embellished with Adobe Illustrator CS6. New sequences generated in this study have been deposited in GenBank (https://www.ncbi.nlm.nih.gov, accessed on 19 March 2024).

## 3. Results

### 3.1. Phylogenetic Analyses

Samples of bamboo plants showing obvious fungal growth were collected from the Baizhu Garden of Fujian Agriculture and Forestry University and West Lake Park of Fuzhou City, Fujian Province, China. A total of eight fungal isolates with different morphological appearances were single-colony purified. For each fungal isolate, ~2637 bp of nucleotide sequences corresponding to portions of the ITS, LSU, *rpb1*, and *tef1-α*loci (ITS: 1–369; LSU: 370–1146; *rpb1*: 1147–1893; *tef1-α*: 1894–2775) were isolated. Based upon initial BLAST results, four of these sequences were isolated, combined with sequences from 68 closely related species as determined by BLAST searches, as well as homologous regions from *Ophioceras dolichostomum* (CBS 114926) and *Ophioceras leptosporum* (CBS 894.70) (Ophioceraceae, Magnaporthales) used as the outgroup for phylogenetic analyses. These analyses showed 1297 distinct patterns, with 1349 bp identical, 614 variable, including gaps, and 812 bp which were parsimony-informative. Maximum likelihood phylogenies were inferred using IQ-TREE under the TIM2 + R4 + F model for 5000 ultrafast bootstraps, as well as the Shimodaira–Hasegawa-like approximate likelihood-ratio test [nst = 6, rates = invgamma], with an average standard deviation of split frequencies = 0.005812. The topological results obtained from the ML analysis were consistent with the results of the BI analysis connecting the combined datasets. As a result, the ML tree is shown, and the BI posterior probabilities are placed on it (Figure 1). Based on phylogenetic resolution and morphological analysis (given below), we report two of the four isolates as new species of Magnaporthaceae: *Bifusisporella fujianensis* and *Bifusisporella bambooensis*. The new species *B. fujianensis* was most closely related to *B. sichuanensis* (SICAUCC 22-0073) (ML-BS: 96%, BYPP: 0.76), and *B. bambooensis* to *B. sorghi* (URM 7442, URM 7864) (ML-BS: 100%, BYPP: 1).

Initial BLAST results of sequences derived from the remaining four isolates indicated placement of these within the Apiosporaceae family. Analyses using sequences derived from the four genetic loci examined, namely the ITS + LSU + *tef1-α* + *tub2* concatenated sequence dataset which had an aligned length of 1941 total characters (ITS: 1–507, LSU: 508–1341, *tub2*: 1342–1830, *tef1-α*: 1831–1941), supported the classification of two of the isolates as new species. Based on these and morphological data (below), a new species, *Apiospora fujianensis,* was identified, related to *A. garethjonesii* (SICAUCC 22-0028), with good support (ML-BS: 93% and BYPP: 0.98). Designation of the other new species, *A. fuzhouensis*, was similarly strongly supported (100% ML/1 PP), with the species forming a separate branch within *Apiospora*. For *A. fuzhouensis* loci analyses, 924 distinct patterns were identified, with 1186 bp constant, 157 variable and included gaps, and 598 bp which were parsimony-informative. Maximum likelihood phylogenies were inferred using IQ-TREE [40], under the GTR + R3 + F model for 5000 ultrafast bootstraps [41], as well as the Shimodaira–Hasegawa-like approximate likelihood-ratio test [nst = 1, rates = invgamma] (Figure 2).

### 3.2. Taxonomy

*Bifusisporella fujianensis* sp. nov. Z.Y. Zhao and J.Z. Qiu, (Figure 3).

MycoBank: MB852815.

Etymology: Named after Fujian Province where the fungus was collected.

Holotype: China, Fujian Province, Fujian University of Agriculture and Forestry (119°14′35.14″ E, 26°5′2.55″ N), from diseased leaves of bamboo in China, March 2023, Z. Y. Zhao (holotype HMAS352712; ex-type living culture CGMCC3.25651).

Description: Leaf spots irregularly shaped, sunken in the center, brown or tan in color. Conidiomata elevated on agar, solitary, spherical, gradually transitioning from white hyaline to black, conidiophores reduced to conidiophores cells. Conidiogenous cells were phialidic, solitary or aggregated, curved, elongated, cylindrical or rod-shaped, light brown, 8.9–14.3 × 5.8–8.1 µm. Conidia were dimorphic, falcate or curved moon-shaped, smooth or cracked surface, transparent in color, 0–3 septa, 37.3–56.3 × 3.6–5.7 µm, mean = 45.1 × 4.4 µm. No sexual morphology was observed.

Culture characteristics: Colonies flattened on PDA with feathery margins, white, on SNA surface and reverse, white. The calculated growth rate was 0.6 cm/day at 25 °C.

Material examined: China, Fujian Province, Fujian University of Agriculture and Forestry (119°14′35.14″ E, 26°5′2.55″ N), from diseased leaves of bamboo in China, March 2023, Z. Y. Zhao (paratype HMAS352713; ex-paratype living culture CGMCC3.27206).

Notes: The strain of the genus *Bifusisporella* was identified as a new species; nucleotide comparison of ITS, LSU, *tef1-α*, and *rpb1* (CGMCC3.25651) showed differences with the sequences of *B. sichuanensis* (SICAUCC 22-0071), similarities are 12.3% (64/522), 4.1% (33/797), 4.1% (36/884), and 8.5% (58/684). In addition, the asexual morph of *B. sichuanensis* was not observed and the sexual morph of *B. fujianensis* was not observed.

*Bifusisporella bambooensis* sp. nov. Z.Y. Zhao and J.Z. Qiu, (Figure 4).

MycoBank: MB852816.

Etymology: The epithet “*bambooensis*” refers to the host, which is bamboo.

Holotype: China, Fujian Province, Fujian University of Agriculture and Forestry, (119°14′35.14″ E, 26°5′2.55″ N), from diseased leaves of bamboo. March 2023, Z.Y. Zhao (holotype HMAS352714; ex-type living culture CGMCC3.25653).

Description: Leaf spots were pike-shaped, color gradually changing from blackish brown to white from outside to inside, Conidiomata bulging on agar, black to hyaline, aggregated and spherical, conidiophores reduced to conidiophores cells. Conidiogenous cells were phialidic, singly or in groups, curved, elongated, cylindrical, 7.2–21.0 × 4.2–6.4 µm, conidia were dimorphic, falcate or curved moon-shaped, smooth or cracked surface, hyaline, 0–3 septa, 10.8–45.0 × 2.8–4.9 µm, mean = 25.0 × 3.7 µm. No sexual morphology was observed.

Culture characteristics: Colonies flattened on PDA, irregular black center, fading to white with white feathery margins, on SNA surface and reverse, white. Calculated growth rate was 1.0–1.4 cm/day at 25 °C. The growth rate was 0.5 cm/day.

Material examined: China, Fujian Province, Fujian University of Agriculture and Forestry (119°14′35.14″ E, 26°5′2.55″ N), from diseased leaves of bamboo in China, March 2023, Z.Y. Zhao (paratype HMAS352715; ex-paratype living culture CGMCC3.27207).

Notes: *Bifusisporella bambooensis* is phylogenetically close (100% ML and 1BYPP), but distinct from *B. sorghi* (URM 7442). Compared to *Bifusisporella sorghi*, *Bifusisporella bambooensis* sp. nov. has larger conidiogenous cells and conidia (7.2–21.0 × 4.2–6.4 vs. 5.0–19.5 × 3.0–4.0 μm; 10.8–45.0 × 2.8–4.9 vs. 19.0–34.0 × 3.0–4.0 μm); nucleotide comparison of ITS, *tef1-α* and *rpb1* (CGMCC3.25653) showed separation from *B. sorghi* (URM 7442), with differences of 6% (29/487), 5.3% (23/437), and 9.4% (65/693), respectively.

*Apiospora fujianensis* sp. nov. Z.Y. Zhao and J.Z. Qiu, (Figure 5).

MycoBank: MB852818.

Etymology: Named after Fujian Province where the fungus was collected.

Holotype: China, Fujian Province, West Lake Park,119°17′47.09″ E,26°5′57.90″ N, from diseased leaves of bamboo in China, October 2022, J.H. Chen (holotype HMAS352716; ex-type living culture CGMCC3.25647).

Description: Leaf spots irregularly shaped, brown or tan in color. Conidiomata on agar were elevated, solitary or aggregated, spherical, black, conidiophores cells were solitary or aggregated, hyaline rounded, 3.5–5.8 × 3.5–5.2 µm. Conidia were rounded or ellipsoidal, contained globular contents, brown, 7.5–17.0 × 5.6–18.2 µm, mean = 14.0 × 11.6 µm. No sexual morphology was observed.

Culture characteristics: Colonies flattened on PDA, fluffy mycelium, black center with white margins over time; calculated growth rate of 1.3 cm/day at 25 °C.

Material examined: China, Fujian Province, West Lake Park, 119°17′47.09″ E, 26°5′57.90″ N, from diseased leaves of bamboo in China, October 2022, J.H. Chen (paratype HMAS352717; ex-paratype living culture CGMCC3.25648).

Notes: In the present study, two strains were obtained from diseased leaves of bamboo and differed from each other with a high degree of statistical support (BYPP = 0.98 and ML-BS = 93%), although overall analyses indicated that both isolates represented different strains of the same species.

*Apiospora fuzhouensis* sp. nov. Z.Y. Zhao and J.Z. Qiu, (Figure 6).

MycoBank: MB852820.

Etymology: Named after Fuzhou, Fujian Province, where the fungus was collected.

Holotype: China, Fujian Province, Fujian University of Agriculture and Forestry, (119°14′35.14″ E, 26°5′2.55″ N), from diseased leaves of bamboo. March 2023, Z.Y. Zhao (holotype HMAS352718; ex-type living culture CGMCC3.25649).

Description: Leaf spots irregular in shape, brown or tan in color. Conidiomata on agar are elevated, solitary or aggregated, spherical, black, Conidiophores hyaline to light brown, smooth, fusiform, subcylindrical, conidiophore cells were solitary or aggregated, hyaline rounded, 1.5–9.1 × 2.4–7.2 µm. Conidia were rounded or ellipsoidal, brownish, 11.3–19.3 × 8.7–19.5 µm, mean = 14.8 × 14.4 µm. No sexual forms were observed.

Culture characteristics: Colonies flattened on PDA; mycelium fluffy, black; calculated growth rate 1.2 cm/day.

Material examined: China, Fujian Province, Fujian University of Agriculture and Forestry, (119°14′35.14″ E, 26°5′2.55″ N), from diseased leaves of bamboo. March 2023, Z.Y. Zhao (paratype HMAS352719; ex-paratype living culture CGMCC3.25650).

Notes: Two strains were obtained from diseased leaves of bamboo and differed from each other with a high degree of statistical support (100% ML/1 PP, Figure 2), although overall analyses indicated that both isolates represented different strains of the same species. The nucleotide comparison of ITS sequences of *A. garethjonesii* (SICAUCC 22-0028) revealed 39 bp (39/542 bp, 7.2%) nucleotide differences. The nucleotide comparison of *tub2* sequences of *A. garethjonesii* (SICAUCC 22-0028) revealed 20 bp (20/524 bp, 3.8%) nucleotide differences. Morphologically, the conidia of *A. fuzhouensis* were slightly smaller than those of *A. garethjonesii* (SICAUCC 22-0028). Therefore, the two strains are proposed as a new species.

## 4. Discussion

As interest in bamboo has intensified due to its wide range of beneficial environmental effects as well as agricultural, industrial, and even foodstuff uses, identification of pathogens, that can decrease quality and/or yield of the plant has also gained interest. Here, we have identified four new species of fungi from diseased bamboo leaves found in Fujian Province, China. Identification was conducted using morphological and molecular phylogenetic analyses, with the former, i.e., characterization of the conidiomata, conidiophores, and conidiogenous cells used as important lines of evidence supporting species identification [42] and the latter (molecular approaches) allowing for phylogenetic placement and confirmation of new species designations.

Two of new species were identified as belonging to the *Bifusisporella* genus. Previously, Silva et al. isolated an endophyte, *Bifusisporella sorgh*, from healthy sorghum leaves in Brazil [15], and another endophyte, *B. sichuanensis*, has been reported from leaves of Sichuan poplar [43]. Most *Bifusisporella* species have sickle-shaped ditype conidia and are commonly found in Poaceae. The newly described species in this report, *B. fujianensis*, grouped with *B. sichuanensis,* but was distinct from the latter in both morphology and multilocus sequence analyses, whereas *B. bambooensis* potentially represents a separate clade. Morphological differences between the species were evident, particularly concerning the conidia (Table 3).

The remaining two new species identified in this report were found to belong to the *Apiospora* genus. Crous and Groenewald synonymized *Apiospora* with *Arthrinium* [44]; however, with additional genetic data from the *Arthrinium* type species, *A. caricicola*, *Apiospora* and *Arthrinium* were separated into two distinct genera [19]. Biogeographically, most specimen of *Arthrinium* have been found in temperate and boreal zones, whereas those of *Apiospora* have been mainly collected in tropical and subtropical regions, with the latter genus displaying a relatively wider distribution area. Currently, therefore, based on the molecular phylogenetic analysis of multigene loci (ITS, LSU, and exon sequences of *tef1-α* and *tub2*), *Arthrinium* and *Apiospora* are considered to represent independent lineages within the Apiosporaceae [19], confirming that the overall genetic, morphological, and ecological differences between *Apiospora* and *Arthrinium* are sufficient to support the taxonomic separation of the two genera. *Apiospora* are characterized by round/lenticular conidia, which are mainly found in Poaceae. Based on morphology and molecular analyses, *Apiospora fujianensis* sp. nov. and *Apiospora fuzhouensis* sp. nov. were described as two new species within *Apiospora.*

As a neo-tropical region, fungal species diversity in Fujian and surrounding areas appears particularly robust [26,27]. However, thus far, only a few species of fungi have been found in bamboo leaves, with our understanding of the diversity of fungal parasitism on bamboo incomplete. This is likely due to bamboo being particularly hardy and resistant to many microorganisms combined with a lack of specimen and data support [45]. These factors necessitate the collection of diverse specimens [5], as well as exploring the reaction/defense by the plant. Here, we provide candidate fungi, with further genomic and physiological studies aimed towards understanding the nature of these fungi on bamboo warranted.

## Figures and Tables

**Figure 1 jof-10-00454-f001:**
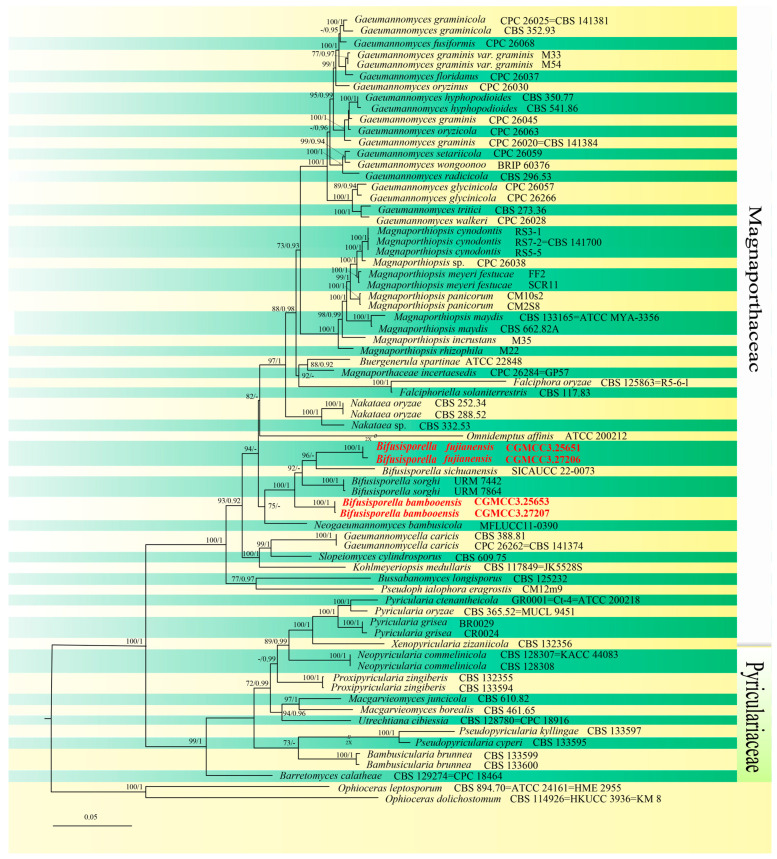
ML tree generated from combined ITS, LSU, *rpb1*, and *tef1-α* sequence data of Magnaporthaceae and Pyriculariaceae. The maximum likelihood (ML) bootstrap support values and Bayesian posterior probabilities (BYPP) bootstrap support values above 70% and 0.90 are shown at the first and second position. Species with sequences obtained in this study are in boldface and newly generated sequences were indicated in red. *Ophioceras dolichostomum* (CBS 114926) and *O.leptosporum* (CBS894.70) (Ophioceraceae) were used as the outgroup. Yellow-green strips represent different neighboring species.

**Figure 2 jof-10-00454-f002:**
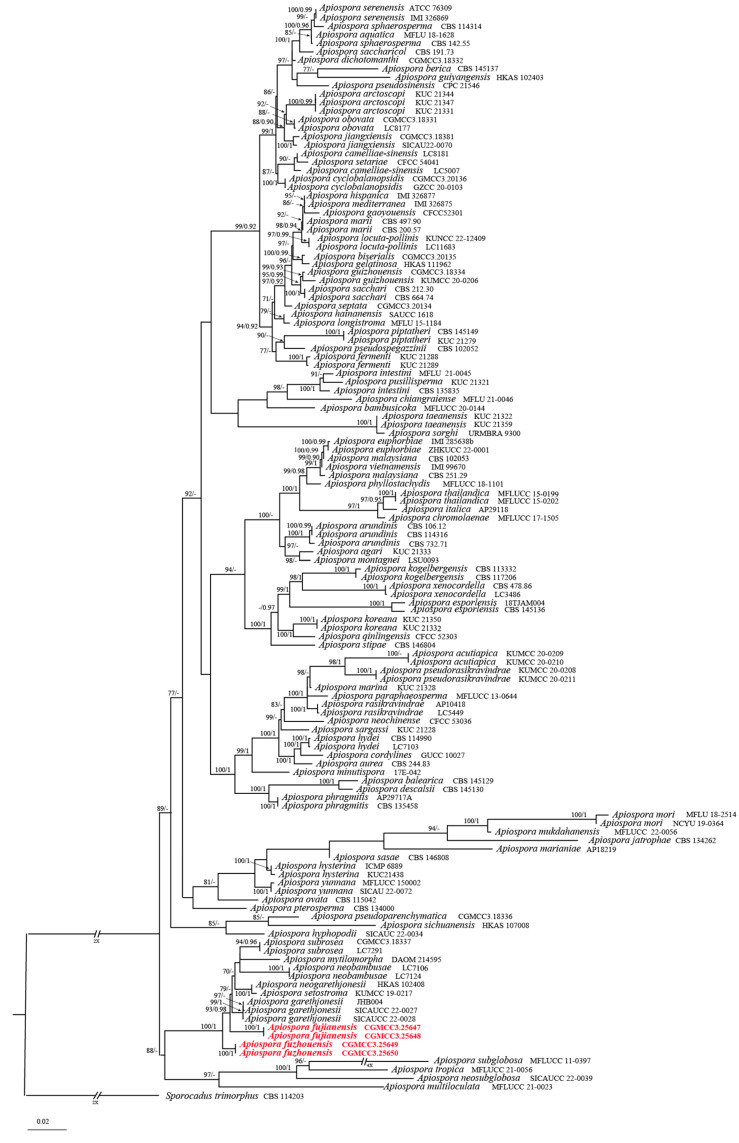
Phylogram of *Apiospora* based on combined ITS, LSU, *tef1-α*, and *tub2* genes. ML bootstrap support values (ML-BS ≥ 70%) and Bayesian posterior probability (BYPP ≥ 0.90) are shown as first and second position above nodes, respectively. Strains from this study are shown in red. Some branches were shortened according to the indicated multipliers.

**Figure 3 jof-10-00454-f003:**
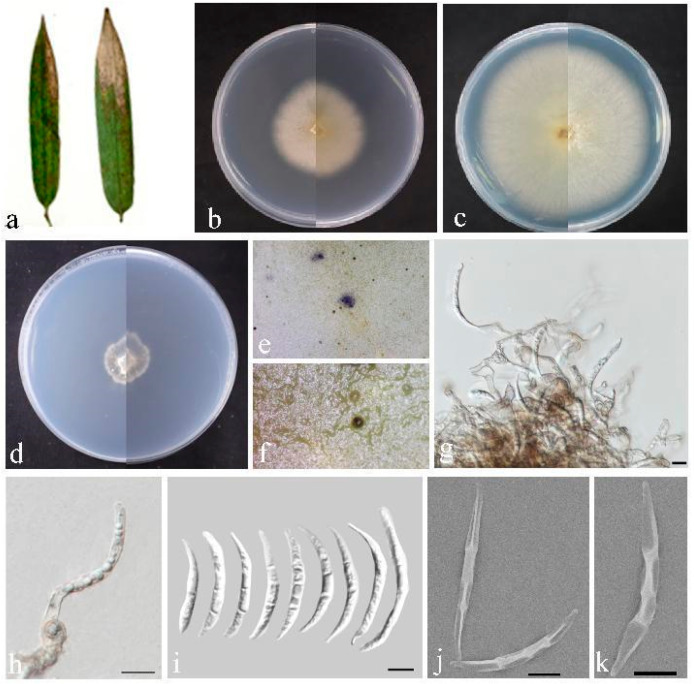
*Bifusisporella fujianensis* (HMAS 352712). (**a**) Leaves of host plant. (**b**,**c**) Upper and reverse view of colony after incubation for 7 days on PDA and 14 days. (**d**) Upper and reverse view of colony after incubation for 14 days on SNA (containing pine needle). (**e**,**f**) Conidiomata sporulating on PDA. (**g**,**h**) Conidiogenous cells and conidia. (**i**–**k**) Conidia. Scalebar = 10 µm (**g**–**k**).

**Figure 4 jof-10-00454-f004:**
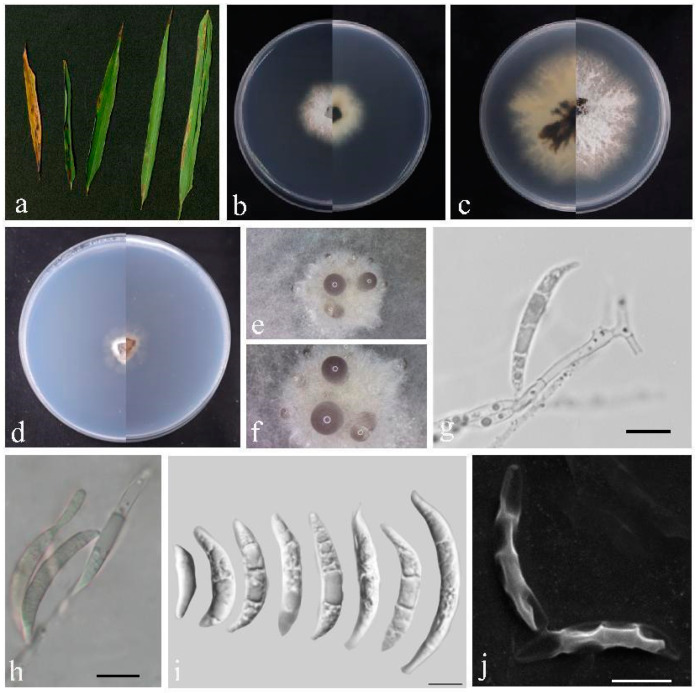
*Bifusisporella bambooensis* (HMAS 352714). (**a**) Leaves of host plant. (**b**,**c**) Upper and reverse view of colony after incubation for 7 days on PDA and 14 days. (**d**) Upper and reverse view of colony after incubation for 14 days on SNA (containing pine needle). (**e**,**f**) Conidiomata sporulating on PDA. (**g**,**h**) Conidiogenous cells and conidia.(**i**,**j**) Conidia. Scalebar = 10 µm (**g**–**j**).

**Figure 5 jof-10-00454-f005:**
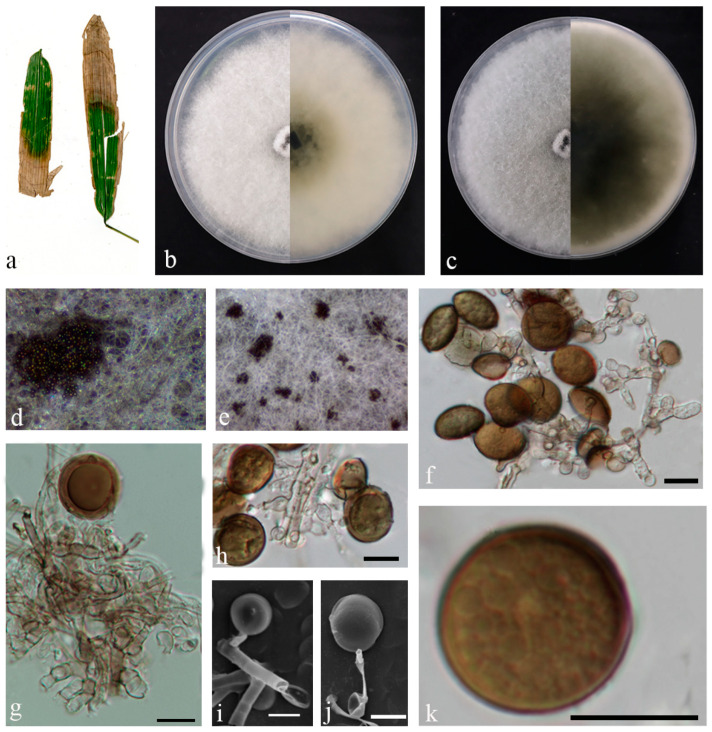
*Apiospora fujianensis* (HMAS 352716). (**a**) Leaves of host plant. (**b**,**c**) Upper and reverse view of colony after incubation for 7 days on PDA and 14 days. (**d**,**e**) Conidiomata sporulating on PDA. (**f**–**j**) Conidiogenous cells and conidia. (**k**) Conidia. Scale bars = 10 µm (**f**–**k**).

**Figure 6 jof-10-00454-f006:**
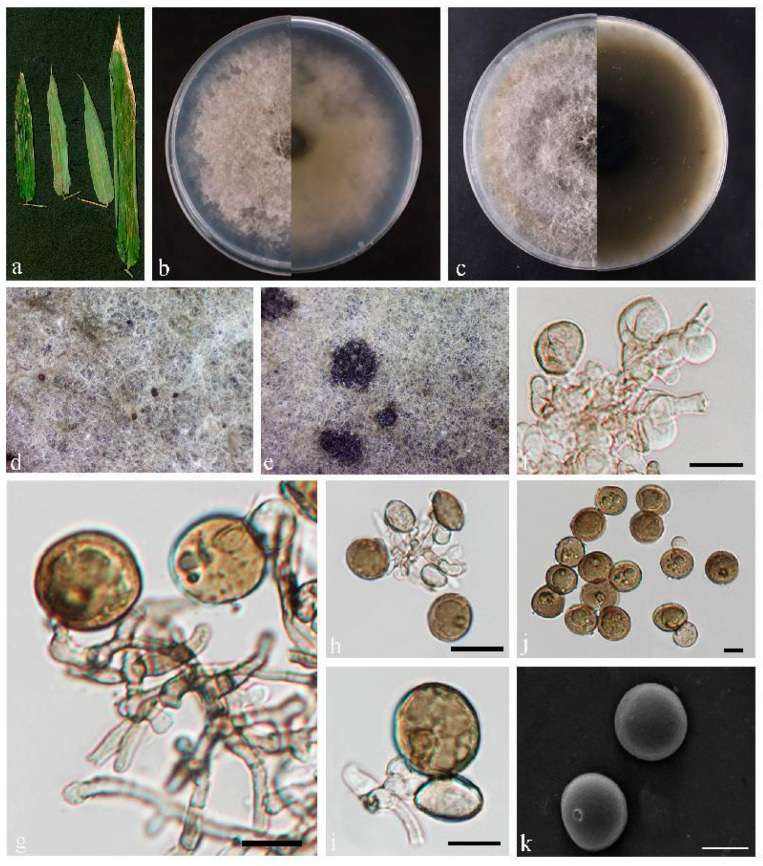
*Apiospora fuzhouensis* (HMAS 352718). (**a**) Leaves of host plant. (**b**,**c**) Upper and reverse view of colony after incubation for 7 days on PDA and 14 days. (**d**,**e**) Conidiomata sporulating on PDA. (**f**–**i**) Conidiogenous cells and conidia. (**j**,**k**) Conidia. Scale bars = 10 µm (**f**–**k**).

**Table 1 jof-10-00454-t001:** The primer sequences and programs in this study.

Locus	Primers	Sequence (5′–3′)	PCR Cycles	References
ITS	ITS5	GGA AGT AAA AGT CGT AAC AAG G	(95 °C: 30 s, 55 °C: 30 s, 72 °C: 1 min) × 35 cycles	[28]
	ITS4	TCCTCCGCTTATTGATATGC		
LSU	LROR	GTACCC GCTGAACTTAAGC	(95 °C: 30 s, 52 °C: 30 s, 72 °C: 1 min) × 35 cycles	[29]
	LR5	TCCTGAGGGAAACTTCG		
*rpb1*	fRPB1-Ac	GAR TGY CCD GGD CAY TTY GG	(95 °C: 30 s, From 57 °C to 72 °C at 0.2 °C/s:30 s, 72 °C: 1 min) × 35 cycles	[30,31]
	fRPB1-Cr	CCNGCDATNTCRTTRTCCATRTA		
*tef1-α*	EF1-983F	GCYCCYGGHCAYCGTGAYTT	(95 °C: 30 s, 57/52 °C: 30 s, 72 °C: 1 min) × 35 cycles	[32]
	EF1-2218R	ATGACACCRACRGCRACRGTYTGYAT		
	EF1-728F	CATCGAGAAGTTCGAGAAGG	(95°C: 30 s, 51 °C: 30 s, 72 °C: 1 min) × 35 cycles	[33]
	EF-2	GGARGTACCAGTSATCATGTT		
*tub2*	Bt2a	GGTAACCAAATCGGT GCTGCT TTC	(95 °C: 30 s, 56 °C: 30 s, 72 °C: 1 min) × 35 cycles	[34]
	Bt2b	ACCCTCAGTGTAGTGACCCTTGGC		

**Table 2 jof-10-00454-t002:** Species names, voucher or culture codes, hosts or substrate, locations, and corresponding GenBank accession numbers of DNA sequences used in this study.

Species	Culture/Voucher	Host/Substrate	Country	GenBank Accession Number				
				ITS	LSU	*tef1-a*	*tub2*	*rpb1*
*Barretomyces calatheae*	CBS 129274 = CPC 18464	*Calathea longifolia*	Brazil	KM484831	KM484950	-	-	KM485045
*Bambusicularia brunnea*	CBS 133599	*Sasa* sp.	Japan	KM484830	KM484948	-	-	KM485043
*Bambusicularia brunnea*	CBS 133600	*Phyllostachys bambusoides*	Japan	AB274436	KM484949	-	-	KM485044
** *Bifusisporella bambooensis* **	**CGMCC3.25653**	***Bambusoideae* sp.**	**China**	**PP159031**	**PP159039**	**PP488459**	-	**PP488463**
** *Bifusisporella bambooensis* **	**CGMCC3.27207**	***Bambusoideae* sp.**	**China**	**PP477445**	**PP477439**	**PP488461**	-	**PP488465**
** *Bifusisporella fujianensis* **	**CGMCC3.25651**	***Bambusoideae* sp.**	**China**	**PP159030**	**PP159038**	**PP488458**	-	**PP488462**
** *Bifusisporella fujianensis* **	**CGMCC3.27206**	***Bambusoideae* sp.**	**China**	**PP477444**	**PP477438**	**PP488460**	-	**PP488464**
*Bifusisporella sorghi*	URM 7442	*Sorghum bicolor* *endophyte*	Brazil	MK060155	MK060153	MK060157	-	MK060159
*Bifusisporella sorghi*	URM 7864	*Sorghum bicolorendophyte*	Brazil	MK060156	MK060154	MK060158	-	MK060160
*Bifusisporella sichuanensis*	SICAUCC 22-0073	*Phyllostachys edulis*	China	ON227097	ON227101	ON244427	-	ON244428
*Bussabanomyces longisporus*	CBS 125232	*Amomum siamense*, leaves	Thailand	KM484832	KM484951	KM009202	-	KM485046
*Buergenerula spartinae*	ATCC 22848	*Spartina alterniflora*, leaves	USA	JX134666	DQ341492	JX134692	-	JX134720
*Falciphora oryzae*	CBS 125863= R5-6-l	*Oryza sativa,* root, endophyte	China	EU636699	KJ026705	JN857963	-	KJ026706
*Falciphoriella solaniterrestris*	CBS 117.83	Soil in potato field	Netherlands	KM484842	KM484959	-	-	KM485058
*Gaeumannomycella graminis*	CPC 26020 = CBS 141384	*Cynodon dactylon* × *C. transvaalensis*	USA	KX306498	KX306568	KX306701	-	KX306633
*Gaeumannomycella graminicola*	CPC 26025 = CBS 141381	*Stenotaphrum secundatum*	USA	KX306495	KX306565	KX306698	-	KX306630
*Gaeumannomycella caricis*	CPC 26262 = CBS 141374	*Carex rostrata*	UK	KX306478	KX306548	KX306675	-	KX306671
*Gaeumannomycella caricis*	CBS 388.81	*Carex rostrata*	UK	KM484843	KM484960	KX306674	-	-
*Gaeumannomyces floridanus*	CPC 26037	*Stenotaphrum secundatum*	USA	KX306491	KX306561	KX306693	-	KX306626
*Gaeumannomyces fusiformis*	CPC 26068	*Oryza sativa*	USA	KX306492	KX306562	KX306694	-	KX306627
*Gaeumannomyces glycinicola*	CPC 26266	*Glycine max*	USA	KX306494	KX306564	KX306696	-	KX306629
*Gaeumannomyces glycinicola*	CPC 26057	*Glycine max*	USA	KX306493	KX306563	KX306695	-	KX306628
*Gaeumannomyces graminicola*	CBS 352.93	-	-	KM484834	DQ341496	KX306697	-	KM485050
*Gaeumannomyces graminis*	CPC 26045	*Cynodon dactylon* × *C. transvaalensis*	-	KX306505	KX306575	KX306708	-	KX306640
*Gaeumannomyces graminis* var. *graminis*	M33	-	-	JF710374	JF414896	JF710411	-	JF710442
*Gaeumannomyces graminis* var. *graminis*	M54	-	-	JF414848	JF414898	JF710419	-	JF710444
*Gaeumannomyces hyphopodioides*	CBS 350.77	*Zea mays*, root	UK	KX306506	KX306576	-	-	-
*Gaeumannomyces hyphopodioides*	CBS 541.86	*Triticum aestivum*, seedling	Germany	KX306507	KX306577	KX306709	-	-
*Gaeumannomyces oryzicola*	CPC 26063	*Oryza sativa*	USA	KX306516	KX306586	KX306717	-	KX306646
*Gaeumannomyces oryzinus*	CPC 26030	*Cynodon dactylon* × *C. transvaalensis*	Bahamas	KX306517	KX306587	KX306718	-	KX306647
*Gaeumannomyces radicicola*	CBS 296.53	-	Canada	KM009170	KM009158	KM009206	-	KM009194
*Gaeumannomyces setariicola*	CPC 26059	*Setaria italica*	South Africa	KX306524	KX306594	KX306725	-	KX306654
*Gaeumannomyces tritici*	CBS 273.36	*Triticum aestivum*	Argentina	KX306525	KX306595	KX306729	-	KX306655
*Gaeumannomyces walkeri*	CPC 26028	*Stenotaphrum secundatum*	USA	KX306543	KX306613	KX306746	-	KX306670
*Gaeumannomyces wongoonoo*	BRIP:60376	*Stenotaphrum secundatum*	Australia	KP162137	KP162146	-	-	-
*Kohlmeyeriopsis medullaris*	CBS 117849 = JK5528S	*Juncus roemerianus*	USA	KM484852	KM484968	-	-	KM485068
*Macgarvieomyces borealis*	CBS 461.65	*Juncus effiisus*, leaf spots	UK	MH858669	DQ341511	KM009198	-	KM485070
*Macgarvieomyces juncicola*	CBS 610.82	*Juncus effiisus*, stem base	The Netherlands	KM484855	KM484970	KM009201	-	KM485071
*Magnaporthaceae,* incertaesedis	CPC 26284 = GP57	*Triticum aestivum*	UK	KX306546	KX306616	KX306677	-	-
*Magnaporthiopsis*sp.	CPC 26038	*Cynodon dactylon* × *C. transvaalensis*	USA	KX306545	-	KX306676	-	KX306672
*Magnaporthiopsis incrustans*	M35	-	-	JF414843	JF414892	-	-	JF710437
*Magnaporthiopsis maydis*	CBS 133165 = ATCC MYA-3356	*Zeamays*	Israel	KX306544	KX306614	-	-	-
*Magnaporthiopsis maydis*	CBS 662.82A	*Zeamays*	Egypt	KM484856	KM484971	-	-	KM485072
*Magnaporthiopsis cynodontis*	RS7-2 = CBS 141700	ultradwarf bermudagrass roots	USA	KJ855508	KM401648	KP282714	-	KP268930
*Magnaporthiopsis cynodontis*	RS5-5	roots	USA	KJ855506	KM401646	KP282712	-	KP268928
*Magnaporthiopsis cynodontis*	RS3-1	roots	USA	KJ855505	KM401645	KP282711	-	KP268927
*Magnaporthiopsis meyeri-festucae*	FF2	-	-	MF178146	MF178151	MF178167	-	MF178162
*Magnaporthiopsis meyeri-festucae*	SCR11	-	-	MF178150	MF178155	MF178171	-	MF178166
*Magnaporthiopsis panicorum*	CM2S8	-	-	KF689643	KF689633	KF689623	-	KF689613
*Magnaporthiopsis panicorum*	CM10s2	-	-	KF689644	KF689634	KF689624	-	KF689614
*Magnaporthiopsis rhizophila*	M22	-	-	JF414833	JF414882	JF710407	-	JF710431
*Nakataea* *sp.*	CBS 332.53	*Oryza sativa*	USA	KM484867	KM484981	-	-	KM485083
*Nakataea oryzae*	CBS 252.34	*Oryza sativa*	Burma	KM484862	KM484976	-	-	KM485078
*Nakataea oryzae*	CBS 288.52	*Oryza sativa*, stem	Japan	KM484864	KM484978	-	-	KM485080
*Neogaeumannomyces bambusicola*	MFLUCC11-0390	Dead culm of bamboo (Bambusae)	Thailand	KP744449	KP744492	-	-	-
*Neopyricularia commelinicola*	CBS 128307 = KACC 44083	*Commelina communis*, leaves	Korea	FJ850125	KM484984	KM009199	-	KM485086
*Neopyricularia commelinicola*	CBS 128308	*Commelina communis*, leaves	Korea	FJ850122	KM484985	-	-	KM485087
*Omnidemptus affinis*	ATCC 200212	*Panicum effiisum* var. effiisum grass leaves	Australia	JX134674	KX134686	JX134700	-	JX134728
*Ophioceras dolichostomum*	CBS 114926 = HKUCC 3936 = KM 8	Wood	China	JX134677	JX134689	JX134703	-	JX134731
*Ophioceras leptosporum*	CBS 894.70 = ATCC 24161 = HME 2955	Dead stem of dicot plant (probably Urtica dioicd)	UK	JX134678	JX134690	JX134704	-	JX134732
*Proxipyricularia zingiberis*	CBS 132355	*Zingiber mioga*	Japan	AB274433	KM484987	-	-	KM485090
*Proxipyricularia zingiberis*	CBS 133594	*Zingiber mioga*	Japan	AB274434	KM484988	-	-	KM485091
*Pseudoph ialophora eragrostis*	CM12m9	*Eragrostis* sp.	USA	KF689648	KF689638	KF689628	-	KF689618
*Pseudopyricularia cyperi*	CBS 133595	*Cyperus iria*	Japan	KM484872	KM484990	-	-	AB818013
*Pseudopyricularia kyllingae*	CBS 133597	*Kyllinga brevifolia*	Japan	KM484876	KM484992	KT950880	-	KM485096
*Pyricularia grisea*	BR0029	*Digitaria sanguinalis*	Brazil	KM484880	KM484995	-	-	KM485100
*Pyricularia grisea*	CR0024	*Lolium perenne*	Korea	KM484882	KM484997	-	-	KM485102
*Pyricularia ctenantheicola*	GR0001 = Ct-4 = ATCC 200218	*Ctenanthe oppenheimiana*	Greece	KM484878	KM484994		-	KM485098
*Pyricularia oryzae*	CBS 365.52 = MUCL 9451	-	Japan	KM484890	KM485000	-	-	KM485110
*Slopeiomyces cylindrosporus*	CBS 609.75	Grass root, associated with Phialophora graminicola	UK	KM484944	KM485040	JX134693	-	KM485158
*Utrechtiana cibiessia*	CBS 128780 = CPC 18916	*Phragmites australis*, leaves	Netherlands	JF951153	JF951176	-	-	KM485047
*Xenopyricularia zizaniicola*	CBS 132356	*Zizania latifolia*	Japan	KM484946	KM485042	KM009203	-	KM485160
*Apiospora acutiapica*	KUMCC 20-0209	-	-	MT946342	MT946338	MT947359	MT947365	-
*Apiospora acutiapica*	KUMCC 20-0210	*Bambusa bambos*	China	-	MT946339	MT947360	MT947366	-
*Apiospora agari*	KUC 21333	*Agarum cribrosum*	Korea	-	MH498440	MH544663	MH498478	-
*Apiospora aquatica*	MFLU 18-1628	Submerged wood	China	MK828608	MK835806	-	-	-
*Apiospora arctoscopi*	KUC 21331	Egg of Arctoscopus japonicus	Korea	-	MH498449	MN868918	MH498487	-
*Apiospora arctoscopi*	KUC 21344	-	-	MH498528	-	MN868919	MH498486	-
*Apiospora arctoscopi*	KUC 21347	-	-	MH498525	-	MN868922	MH498483	-
*Apiospora arundinis*	CBS 114316	*Hordeum vulgare*	Iran	KF144884	KF144928	KF145016	KF144974	-
*Apiospora arundinis*	CBS 106.12	-	-	KF144883	KF144927	KF145015	KF144973	-
*Apiospora arundinis*	CBS 732.71	-	-	KF144889	KF144934	KF145022	KF144980	-
*Apiospora aurea*	CBS 244.83	Air	Spain	AB220251	KF144935	KF145023	KF144981	-
*Apiospora balearica*	CBS 145129	Poaceae	Spain	MK014869	MK014836	MK017946	MK017975	-
*Apiospora neobambusae*	HMAS LC7106	-	-	KY494718	KY494794	KY806204	KY705186	-
*Apiospora bambusicola*	MFLUCC20-0144	*Schizostachyum brachycladum*	Thailand	MW173030	MW173087	MW183262	-	-
*Apiospora biserialis*	CGMCC 3.20135	*Bambusoideae* sp.	China	MW481708	MW478885	MW522938	MW522955	-
*Apiospora camelliae-sinensis*	LC5007	*Camellia sinensis*	China	KY494704	KY494780	KY705103	KY705173	-
*Apiospora camelliae-sinensis*	LC8181	-	-	KY494761	KY494837	KY705157	KY705229	-
*Apiospora chiangraiense*	MFLU 21-0046	-	-	MZ542520	MZ542524	-	MZ546409	-
*Apiospora chromolaenae*	MFLUCC 17-1505	*Chromolaena odorata*	Thailand	MT214342	MT214436	MT235802	-	-
*Apiospora cordylines*	GUCC 10027	-	-	MT040106	-	MT040127	MT040148	-
*Apiospora cyclobalanopsidis*	CGMCC 3.20136	*Cyclobalanopsis glauca*	China	MW481713	MW478892	MW522945	MW522962	-
*Apiospora cyclobalanopsidis*	GZCC:20-0103	-	-	MW481714	-	MW522946	MW522963	-
*Apiospora descalsii*	CBS 145130	*Ampelodesmos mauritanicus*	Spain	MK014870	MK014837	MK017947	MK017976	-
*Apiospora dichotomanthi*	CGMCC 3.18332	*Dichotomanthes tristaniiaecarpa*	China	KY494697	KY494773	KY705096	KY705167	-
*Apiospora esporlensis*	CBS 145136	*Phyllostachys aurea*	Spain	MK014878	MK014845	MK017954	MK017983	-
*Apiospora esporlensis*	18TJAM004	-	-	MT856406	-	MT881953	MT881991	-
*Apiospora euphorbiae*	IMI 285638b	*Bambusoideae* sp.	Bangladesh	AB220241	AB220335	-	AB220288	-
*Apiospora euphorbiae*	ZHKUCC 22-0001	-	-	OM728647	OM486971	OM543543	OM543544	-
*Apiospora fermenti*	KUC 21289	-	-	MF615226	-	MH544667	MF615231	-
*Apiospora fermenti*	KUC 21288	-	-	MF615230	-	MH544668	MF615235	-
** *Apiospora fujianensis* **	**CGMCC3.25647**	***Bambusoideae* sp.**	**China**	**PP159026**	**PP159034**	**PP488454**	**PP488470**	**-**
** *Apiospora fujianensis* **	**CGMCC3.25648**	***Bambusoideae* sp.**	**China**	**PP159027**	**PP159035**	**PP488455**	**PP488471**	**-**
** *Apiospora fuzhouensis* **	**CGMCC3.25649**	***Bambusoideae* sp.**	**China**	**PP159028**	**PP159036**	**PP488456**	**PP488468**	**-**
** *Apiospora fuzhouensis* **	**CGMCC3.25650**	***Bambusoideae* sp.**	**China**	**PP159029**	**PP159037**	**PP488457**	**PP488469**	**-**
*Apiospora gaoyouensis*	CFCC52301	*Phragmites australis*	China	MH197124	-	MH236793	MH236789	-
*Apiospora garethjonesii*	JHB004	Culms of dead bamboo	China	KY356086	KY356091	-	-	-
*Apiospora garethjonesii*	SICAUCC 22-0028	-	-	ON228606	ON228662	-	ON237654	-
*Apiospora garethjonesii*	SICAUCC 22-0027	-	-	ON228603	ON228659	-	ON237651	-
*Apiospora gelatinosa*	HKAS 111962	Culms of dead bamboo	China	MW481706	MW478888	MW522941	MW522958	-
*Apiospora guiyangensis*	HKAS 102403	Dead culms of Poaceae	China	MW240647	MW240577	MW759535	MW775604	-
*Apiospora guizhouensis*	CGMCC 3.18334	Air in karst cave	China	KY494709	KY494785	KY705108	KY705178	-
*Apiospora guizhouensis*	KUMCC 20-0206	-	-	MT946347	MT946341	MT947364	MT947370	-
*Apiospora hainanensis*	SAUCC 1681	Leaf of bamboo	China	OP563373	OP572422	OP573262	OP573268	-
*Apiospora hispanica*	IMI 326877	*Maritime sand*	Spain	AB220242	AB220336	-	AB220289	-
*Apiospora hydei*	CBS 114990	*Bambusoideae* sp.	China	KF144890	KF144936	KF145024	KF144982	-
*Apiospora hydei*	LC 7103	-	-	KY494715	KY494791	KY705114	KY705183	-
*Apiospora hyphopodii*	SICAUCC 22-0034	-	-	ON228605	ON228661	-	ON237653	-
*Apiospora hysterina*	ICMP 6889	*Bambusoideae* sp.	New Zealand	MK014874	MK014841	MK017951	MK017980	-
*Apiospora hysterina*	KUC21438	-	-	ON764019	ON787758	ON806623	ON806633	-
*Apiospora iberica*	CBS 145137	*Arundo donax*	Portugal	MK014879	MK014846	MK017955	MK017984	-
*Apiospora intestini*	CBS 135835	Gut of grasshopper	India	KR011352	KR149063	KR011351	KR011350	-
*Apiospora intestini*	MFLU:21-0045	-	-	MZ542521	MZ542525	MZ546406	MZ546410	-
*Apiospora italica*	AP29118	-	-	MK014881	MK014848	MK017957	MK017986	-
*Apiospora jatrophae*	CBS 134262	*Jatropha podagrica*	India	JQ246355	-	-	-	-
*Apiospora jiangxiensis*	CGMCC 3.18381	*Maesa* sp.	China	KY494693	KY494769	KY705092	KY705163	-
*Apiospora jiangxiensis*	SICAU 22-0070	-	-	ON227094	ON227098	ON244431	ON244432	-
*Apiospora kogelbergensis*	CBS 113332	*Cannomois virgata*	South Africa	KF144891	KF144937	KF145025	KF144983	-
*Apiospora kogelbergensis*	CBS 117206	-	-	KF144895	KF144941	KF145029	KF144987	-
*Apiospora koreana*	KUC 21332	Egg of *Arctoscopus japonicus*	Korea	MH498524	-	MH544664	MH498482	-
*Apiospora koreana*	KUC21350	-	-	MH498521	-	MN868929	MH498479	-
*Apiospora locuta-pollinis*	LC11683	*Brassica campestris*	China	MF939595	-	MF939616	MF939622	-
*Apiospora locuta-pollinis*	KUNCC:22-12409	-	-	OP377737	OP377744	OP381091	-	-
*Apiospora longistroma*	MFLU 15-1184	Culms of decaying bamboo	Thailand	KU940141	KU863129	-	-	-
*Apiospora malaysiana*	CBS 102053	*Macaranga hullettii*	Malaysia	KF144896	KF144942	KF145030	KF144988	-
*Apiospora malaysiana*	CBS:251.29	-	-	KF144897	KF144943	KF145031	KF144989	-
*Apiospora marianiae*	AP18219	Dead stems of *Phleum pratense*	Spain	ON692406	ON692422	ON677180	ON677186	-
*Apiospora marii*	CBS 497.90	Air	Spain	MH873913	KF144947	KF145035	KF144993	-
*Apiospora marii*	CBS 200.57	-	-	KF144900	KF144946	KF145034	KF144992	-
*Apiospora marina*	KUC 21328	Seaweed	Korea	MH498538	MH498458	MH544669	MH498496	-
*Apiospora mediterranea*	IMI 326875	Air	Spain	AB220243	AB220337	-	AB220290	-
*Apiospora minutispora*	17E 042	Soil	Korea	LC517882	-	LC518889	LC518888	-
*Apiospora montagnei*	LSU0093	-	-	MT000394	MT000490	-	-	-
*Apiospora mori*	MFLU 18-2514	Dead leaves of *Morus australis*	China	MW114313	MW114393	-	-	-
*Apiospora mori*	NCYU 19-0364	-	-	MW114314	MW114394	-	-	-
*Apiospora mukdahanensis*	MFLUCC 22-0056	-	-	OP377735	OP377742	OP381089	-	-
*Apiospora multiloculata*	MFLUCC 21-0023	Dead culms of Bambusae	Thailand	OL873137	OL873138	-	OL874718	-
*Apiospora mytilomorpha*	DAOM 214595	Dead blades of Andropogon sp.	India	KY494685	-	-	-	-
*Apiospora neobambusae*	LC7124	-	-	KY494727	KY494803	KY806206	KY705195	-
*Apiospora neochinense*	CFCC 53036	*Fargesia qinlingensis*	China	MK819291	-	MK818545	MK818547	-
*Apiospora neogarethjonesii*	HKAS 102408	Dead culms of Bambusae	China	MK070897	MK070898	-	-	-
*Apiospora neosubglobosa*	SICAUCC 22-0039	-	-	ON228614	ON228670	-	ON237662	-
*Apiospora obovata*	CGMCC 3.18331	*Lithocarpus* sp.	China	KY494696	KY494772	KY705095	KY705166	-
*Apiospora obovata*	LC8177	-	-	KY494757	KY494833	KY705153	KY705225	-
*Apiospora ovata*	CBS 115042	*Arundinaria hindsii*	China	KF144903	KF144950	KF145037	KF144995	-
*Apiospora paraphaeosperma*	MFLUCC13-0644	Dead clumps of Bambusa sp.	Thailand	KX822128	KX822124	-	-	-
*Apiospora phragmitis*	CBS 135458	*Phragmites australis*	Italy	KF144909	KF144956	KF145043	KF145001	-
*Apiospora phragmitis*	AP29717A	-	-	MK014892	MK014859	MK017968	MK017997	-
*Apiospora phyllostachydis*	MFLUCC 18-1101	*Phyllostachys heteroclada*	China	MK351842	MH368077	MK340918	MK291949	-
*Apiospora piptatheri*	CBS 145149	*Piptatherum miliaceum*	Spain	MK014893	MK014860	-	-	-
*Apiospora piptatheri*	KUC21279	-	-	MF615229	-	MH544671	MF615234	-
*Apiospora pseudoparenchymatica*	CGMCC 3.18336	Bambusoideae sp.	China	KY494743	-	KY705139	KY705211	-
*Apiospora pseudorasikravindrae*	KUMCC 20-0208	-	-	MT946344	-	MT947361	MT947367	-
*Apiospora pseudorasikravindrae*	KUMCC 20-0211	-	-	MT946345	-	MT947362	MT947368	-
*Apiospora pseudosinensis*	CPC 21546	Leaf of bamboo	The Netherlands	KF144910	KF144957	KF145044	MN868936	-
*Apiospora pseudospegazzinii*	CBS 102052	*Macaranga hullettii*	Malaysia	KF144911	KF144958	KF145045	KF145002	-
*Apiospora pterosperma*	CBS 134000	*Machaerina sinclairii*	Australia	KF144913	KF144960	KF145046	KF145004	-
*Apiospora pusillisperma*	KUC 21321	Seaweed	Korea	MH498533	MH498453	MN868930	MH498491	-
*Apiospora qinlingensis*	CFCC 52303	*Fargesia qinlingensis*	China	MH197120	-	MH236795	MH236791	-
*Apiospora rasikravindrae*	LC5449	Soil in karst cave	China	KY494713	KY494789	KY705112	KY705182	-
*Apiospora rasikravindrae*	AP10418	-	-	MK014896	MK014863	-	MK017999	-
*Apiospora sacchari*	CBS 212.30	*Phragmites australis*	UK	KF144916	KF144962	KF145047	KF145005	-
*Apiospora sacchari*	CBS:664.74	-	-	KF144919	KF144965	KF145050	KF145008	-
*Apiospora saccharicola*	CBS 191.73	Air	The Netherlands	KF144920	KF144966	KF145051	KF145009	-
*Apiospora sargassi*	KUC 21228	*Sargassum fulvellum*	Korea	KT207746	-	MH544677	KT207644	-
*Apiospora sasae*	CBS 146808	Dead culms of *Sasa veitchii*	The Netherlands	MW883402	MW883797	MW890104	MW890120	-
*Apiospora septata*	CGMCC 3.20134	*Bambusoideae* sp.	China	MW481711	MW478890	MW522943	MW522960	-
*Apiospora serenensis*	IMI 326869	-	Spain	AB220250	AB220344	-	AB220297	-
*Apiospora serenensis*	ATCC 76309	-	-	AB220240	AB220334	-	AB220287	-
*Apiospora setariae*	CFCC 54041	Decaying culms of *Setaria viridis*	China	MT492004	-	-	-	-
*Apiospora setostroma*	KUMCC 19-0217	Dead branches of bamboo	China	MN528012	MN528011	MN527357	-	-
*Apiospora sichuanensis*	HKAS 107008	Dead culms of Poaceae	China	MW240648	MW240578	MW759536	MW775605	-
*Apiospora sorghi*	URMBRA 9300	*Sorghum bicolor*	Brazil	MK371706	-	-	MK348526	-
*Apiospora sphaerosperma*	CBS114314	Leaf of Hordeum vulgare	Iran	KF144904	KF144951	KF145038	KF144996	-
*Apiospora sphaerosperma*	CBS 142.55	-	-	KF144908	KF144955	KF145042	AB220303	-
*Apiospora stipae*	CBS 146804	*Stipa gigantea*	Spain	MW883403.1	MW883798.1	-	MW890121.1	-
*Apiospora subglobosa*	MFLUCC 11-0397	Dead culms of bamboo	Thailand	KR069112	KR069113	-	-	-
*Apiospora subrosea*	CGMCC 3.18337	Bambusoideae sp.	China	KY494752	KY494828	KY705148	KY705220	-
*Apiospora subrosea*	LC 7291	-	-	KY494751	KY494827	KY705147	KY705219	-
*Apiospora taeanensis*	KUC 21322	Seaweed	Korea	MH498515	-	MH544662	MH498473	-
*Apiospora taeanensis*	KUC 21359	-	-	MH498513	-	MN868935	MH498471	-
*Apiospora thailandica*	MFLUCC 15-0202	-	-	KU940145	KU863133	-	-	-
*Apiospora thailandica*	MFLUCC 15-0199	-	-	KU940146	KU863134	-	-	-
*Apiospora tropica*	MFLUCC 21–0056	-	-	OK491657	OK491653	-	OK560922	-
*Apiospora vietnamensis*	IMI 99670	*Citrus sinensis*	Vietnam	KX986096	KX986111	-	KY019466	-
*Apiospora xenocordella*	CBS 478.86	Soil from roadway	Zimbabwe	KF144925	KF144970	KF145055	KF145013	-
*Apiospora xenocordella*	LC3486	-	-	KY494687	KY494763	KY705086	KY705158	-
*Apiospora yunnana*	MFLUCC 150002	Culms of Decaying bamboo	China	KU940147	KU863135	-	-	-
*Apiospora yunnana*	SICAU 22-0072	-	-	ON227096	ON227100	ON244425	ON244426	-

Notes: newly generated sequences are in bold.

**Table 3 jof-10-00454-t003:** The location, hosts or substrate, and main morphological characters of *Bifusisporella.*

Species	Location	Host/Substrate	Conidiogenous Cells	Size of Conidiophore Cells (µm)	Conidia	Size of Conidia (µm)	References
*Bifusisporella bambooensis* sp. nov.	China	Bambusoideae sp.	Cylindrical	7.2–21.0 × 4.2–6.4	falcate or curved moon-shaped	10.8–45.0 × 2.8–4.9	In this study
*Bifusisporella sorghi*	Brazil	*Sorghum bicolor*	Cylindrical orclavate	5.0–19.5 × 3.0–4.0	falcate	Macroconidia 19.0–34.0 × 3.0–4.0 Microconidia 7.0–14.5 × 1.0–2.0	[15]
*Bifusisporella fujianensis* sp. nov.	China	*Bambusoideae* sp.	Cylindricalor rod-shaped	8.9–14.3 × 5.8–8.1	falcate or curved moon-shaped	37.3–56.3 × 3.6–5.7	In this study
*Bifusisporella sichuanensis*	China	*Phyllostachys edulis*	-	-	-	-	[43]

## Data Availability

All newly generated sequences were deposited in GenBank (https://www.ncbi.nlm.nih.gov/genbank/ (accessed on 19 March 2024). All new taxa were linked with MycoBank (https://www.mycobank.org/ (accessed on 13 March 2024)).

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
