# Peer review of "Diversity and New Species of Ascomycota from Bamboo in China"

_jof, 2024, doi:10.3390/jof10070454_

Round 1
Reviewer 1 Report
The manuscript titled “Diversity and new species of Ascomycota from bamboo in China” is devoted to diversity of fungal species isolated in China as potential pathogens of bamboo. New species of potentially phytopathogenic fungi were isolated from diseased leaves collected within Fujian province and identified based on their morphological characteristics and multilocus phylogenies using nucleotide sequences derived from combined datasets of internal transcribed spacers (ITS), large subunit nuclear rDNA (LSU), the large subunit of RNA polymerase I (rpb1), translation elongation factor1-alpha (tef1-α) and β-tubulin gene 2 (tub2). Authors identified four new species of bambusicolous fungi, consisting of two species of Bifusisporella, described as B. fujianensis sp. nov. and B. bambooensis sp. nov., and two new species of Apiospora, described as A. fujianensis sp. nov. and A. fuzhouensis sp. nov.
The manuscript is well-written and research data look reliable.
Unfortunately, identification of the new species of fungi from bamboo in China as plant pathogens (see the Line 22: “Here, we describe new species of plant pathogenic fungi from the leaves of bamboo within Fujian provinces, China”) is not supported by experiments that fulfill the Koch's postulates. Authors must correct it through the manuscript and write about plant-associated species, not about phytopathogenic ones. Additional tests of pathogenicity must be done using proper laboratory and field experiments.
Besides this, it is recommended to improve phylogenetic trees shown at Line 190 (Figure 1. RAxML tree generated from combined ITS, LSU, rpb2…) and at Line 200 (Figure 2. Phylogram of Apiospora based on combined ITS, LSU, tef1-α and tub2 genes). ML bootstrap support values and Bayesian posterior probability are not read well on those figures. It is possible to skip Bayesian posterior probability without significant loss of information.
Yellow-green strips on Fig. 1 must be explained in the legend of this figure, and green background of the Fig. 2 can be removed for more clear view of the tree.
1) It is recommended to improve phylogenetic trees shown at Line 190 (Figure 1. RAxML tree generated from combined ITS, LSU, rpb2…) and at Line 200 (Figure 2. Phylogram of Apiospora based on combined ITS, LSU, tef1-α and tub2 genes). ML bootstrap support values and Bayesian posterior probability are not read well on those figures. It is possible to skip Bayesian posterior probability without significant loss of information.
2) Yellow-green strips on Fig. 1 must be explained in the legend of this figure,
3) Green background of the Fig. 2 can be removed for more clear view of the tree.
Author Response
Dear Editors and Reviewers:
Thank you for your letter and comments relating to our manuscript entitled “Diversity and new species of Ascomycota from bamboo in China” (ID: jof-3024840). The comments were very helpful in revising and improving our manuscript as well as emphasizing the significance to our research. We have read the comments carefully and made corrections accordingly. Revised portions are marked in blue in the manuscript. The main corrections in the paper and our responses to the reviewer’s comments are given below. We hope that the revisions in the manuscript and our accompanying responses will be sufficient to make our manuscript suitable for publication in the Journal of Fungi.
Responses to the comments of the reviewer:
Reviewer 1#
Comments 1: Unfortunately, identification of the new species of fungi from bamboo in China as plant pathogens (see the Line 22: “Here, we describe new species of plant pathogenic fungi from the leaves of bamboo within Fujian provinces, China”) is not supported by experiments that fulfill the Koch's postulates. Authors must correct it through the manuscript and write about plant-associated species, not about phytopathogenic ones. Additional tests of pathogenicity must be done using proper laboratory and field experiments.
Response 1: We have changed “plant pathogen species” to “plant-associated species”.
Comments 2: Besides this, it is recommended to improve phylogenetic trees shown at Line 190 (Figure 1. RAxML tree generated from combined ITS, LSU, rpb2…) and at Line 200 (Figure 2. Phylogram of Apiospora based on combined ITS, LSU, tef1-α and tub2 genes). ML bootstrap support values and Bayesian posterior probability are not read well on those figures. It is possible to skip Bayesian posterior probability without significant loss of information.
Response 2: Thank you very much for your advice! We have revised it. The poorly read Bayesian posterior in those figures has been skipped to allow for better reading.
Comments 3: Yellow-green strips on Fig. 1 must be explained in the legend of this figure, and green background of the Fig. 2 can be removed for more clear view of the tree.
Response 3: We have added explanation of Yellow-green strips to the legend of Fig. 1 and removed green background of Fig. 2.
Yellow-green strips represent different neighboring species.
Comments 4: It is recommended to improve phylogenetic trees shown at Line 190 (Figure 1. RAxML tree generated from combined ITS, LSU, rpb2…) and at Line 200 (Figure 2. Phylogram of Apiospora based on combined ITS, LSU, tef1-α and tub2 genes). ML bootstrap support values and Bayesian posterior probability are not read well on those figures. It is possible to skip Bayesian posterior probability without significant loss of information.
Response 4: Thank you very much for your advice! We have revised it. The poorly read Bayesian posterior in those figures has been skipped to allow for better reading.
Comments 5: Yellow-green strips on Fig. 1 must be explained in the legend of this figure,
Response 5: We have revised it.
Yellow-green strips represent different neighboring species.
Comments 6: Green background of the Fig. 2 can be removed for more clear view of the tree.
Response 6: We have revised it.
We tried our best to improve the manuscript and made some changes marked in blue in revised paper which will not influence the content and framework of the paper. We appreciate for Editors/Reviewers’ warm work earnestly and hope the revision will meet with your approval. Once again, thank you very much for your comments and suggestions.
Kind regards,
Junzhi Qiu
E-mail address: junzhiqiu@126.com
Reviewer 2 Report
The paper entitled 'Diversity and new species of Ascomycota from bamboo in China' is devoted to identification of four new species from bamboo plants using a polyphasic approach. The paper is written by a classic way and include multilocus phylogeny and morphological characterization of fungal isolates. This works looks timely and woukld be interesting for the specialists in plant pathology and molecular taxonomy. In general, the paper can be published in JoF, but I would like the authors to take into account several comments:
1. I think it would be interesting if, besides concatenated sequence dataset, individual marker genes were tested for their phylogenetic characteristics to distunguish Apiospora and Bifusisporella species. This information seems to be scientifically sound in terms of phylogenetic utility of each of marker tested and their potential to use as barcodes for tested taxons. Also, the key phylogenetic parameters (number of variable, parsimony informative sites etc.) could be combined in a separate table. Moreover in 'Materials and Methods' the authors declare that they use five different loci for phylogentic studies. However, in 'Results' we can see that only four markers were used for phylogenetic studies of Apiospora and Bifusisporella (ITS-LSU-rpb1-TEF1a and ITS-LSU-btub-TEF1a, respectively). I believe the authors have to explain why different set of markers were used. Also, in Fig. 1 and Table S1, rpb2 gene is mentioned. But I can't find any information about its use in the paper's text. These moments must be improved.
2. I can't understand from the paper's text how many bamboo samples were tested and how many isolates were isolated from them. The authors declared that four novel species were described; at the same time, in 'Results' they report about 'a total of four fungal isolates...'. On the other hand, in Table 1 a plenty of isolates including eight isolates of new species are presented. What is true?
3. More information of diversity of Apiospora and Bifusisporella, intra- and interspecific polymorphism of species, including bambusicolous ones would be appropriate for this paper. Also, some interesting references, such as review paper by Jiang et al, 2022 (Studies in Fungi) are missing.
4. I think the authors should check the English grammar and style, probably using any proofreading service. There are some sentences which look hard to understand.
Line 22: '....potential pathogens of bamboo pathogenic...' - 'pathogenic should be removed
Line 28: ..helped TO clarify....
Line 36: Latin name of bamboo?
Line 118-119: Information on RPB2 primers and the reference for b-tub primers are missed.
Line 154: Four isolates or four species?
Line 193 (Fig 1): '.', not ','
Line 296: Why 'RPB2' in capital letters and 'rpb1' and 'rpb2' in Table S1 in lower case?
Line 485: I believe Table S1 should be in the text, not in Supplementary.
Author Response
Dear Editors and Reviewers:
Thank you for your letter and comments relating to our manuscript entitled “Diversity and new species of Ascomycota from bamboo in China” (ID: jof-3024840). The comments were very helpful in revising and improving our manuscript as well as emphasizing the significance to our research. We have read the comments carefully and made corrections accordingly. Revised portions are marked in blue in the manuscript. The main corrections in the paper and our responses to the reviewer’s comments are given below. We hope that the revisions in the manuscript and our accompanying responses will be sufficient to make our manuscript suitable for publication in the Journal of Fungi.
Responses to the comments of the reviewer:
Reviewer 2#
Comments 1: I think it would be interesting if, besides concatenated sequence dataset, individual marker genes were tested for their phylogenetic characteristics to distunguish Apiospora and Bifusisporella species. This information seems to be scientifically sound in terms of phylogenetic utility of each of marker tested and their potential to use as barcodes for tested taxons. Also, the key phylogenetic parameters (number of variable, parsimony informative sites etc.) could be combined in a separate table. Moreover in 'Materials and Methods' the authors declare that they use five different loci for phylogentic studies. However, in 'Results' we can see that only four markers were used for phylogenetic studies of Apiospora and Bifusisporella (ITS-LSU-rpb1-TEF1a and ITS-LSU-btub-TEF1a, respectively). I believe the authors have to explain why different set of markers were used. Also, in Fig. 1 and Table S1, rpb2 gene is mentioned. But I can't find any information about its use in the paper's text. These moments must be improved.
Response 1: We have revised them. Based on the following literature and many other related references, we selected ITS-LSU-rpb1-tef1-α as the maker for Bifusisporella and ITS-LSU-tub2-tef1-α as the maker for Apiospora. Each of these two makers can be better used to demonstrate the characteristics of the two genera. For the rpb2 gene portion of the article, we have corrected it.
- Zeng, Q.; Lv, Y.; Xu, X.L.; Deng, Y.; Wang, F.H.; Liu, S.Y.; Liu, L.J.; Yang, C.J.; Liu, Y.G. Morpho-Molecular Characterization of Microfungi Associated with Phyllostachys (Poaceae) in Sichuan, China. J. Fungi 2022, 8, 702. https://doi.org/10.3390/jof8070702.
- Liu, R.; Li, D.H.; Zhang, Z.X.; Liu, S.B.; Liu, X.Y.; Wang, Y.X.; Zhao, H.; Liu, X.Y.; Zhang, X.G.; Xia, J.W.; et al. Morphological and phylogenetic analyses reveal two new species and a new record of Apiospora (Amphisphaeriales, Apiosporaceae) in China. MycoKeys 2023, 95, 27–45. https://doi.org/10.3897/mycokeys.95.96400.
Comments 2: I can't understand from the paper's text how many bamboo samples were tested and how many isolates were isolated from them. The authors declared that four novel species were described; at the same time, in 'Results' they report about 'a total of four fungal isolates...'. On the other hand, in Table 1 a plenty of isolates including eight isolates of new species are presented. What is true?
Response 2: We found two new species with four isolates in Apiospora and two new species with four isolates in Bifusisporella. In total, there are four new species and eight isolates, and some of the expressions in the article have been modified.
Comments 3: More information of diversity of Apiospora and Bifusisporella, intra- and interspecific polymorphism of species, including bambusicolous ones would be appropriate for this paper. Also, some interesting references, such as review paper by Jiang et al, 2022 (Studies in Fungi) are missing.
Response 3: We have revised it and added the interesting reference Jiang et al, 2022.
Comments 4: I think the authors should check the English grammar and style, probably using any proofreading service. There are some sentences which look hard to understand.
Response 4: In order to improve the language, we have asked the native english-speaking author (Prof. Nemat) of this article to polish the English grammar.
Comments 5: Line 22: '....potential pathogens of bamboo pathogenic...' - 'pathogenic should be removed
Response 5: We have revised it.
Comments 6: Line 28: ..helped TO clarify....
Response 6: We have revised it.
Comments 7: Line 36: Latin name of bamboo?
Response 7: Bamboo belongs to the family Bambusoideae, order Poales.
Comments 8: Line 118-119: Information on RPB2 primers and the reference for b-tub primers are missed.
Response 8: We have added them to the revised manuscript.
Comments 9: Line 154: Four isolates or four species?
Response 9: We have revised it. We have eight isolates of four new species.
Comments 10: Line 193 (Fig 1): '.', not ','
Response 10: We have revised it.
Comments 11: Line 296: Why 'RPB2' in capital letters and 'rpb1' and 'rpb2' in Table S1 in lower case?
Response 11: We have revised it.
Comments 12: Line 485: I believe Table S1 should be in the text, not in Supplementary.
Response 12: We have moved Table S1 into the text.
We tried our best to improve the manuscript and made some changes marked in blue in revised paper which will not influence the content and framework of the paper. We appreciate for Editors/Reviewers’ warm work earnestly and hope the revision will meet with your approval. Once again, thank you very much for your comments and suggestions.
Kind regards,
Junzhi Qiu
E-mail address: junzhiqiu@126.com
Reviewer 3 Report
The manuscript is well prepared in most places but there are two places that need some grammar clarification (see attachment). The two major issues are 1) the authors reference Bifusisporella sichuanensis but it is not on the tree, and it appears that index fungorium has more Apiospora species then the authors mention that are on their tree. Since there are only a few Bifusisporella species it would be best to have the B. sichuanensis on the phylogenetic reconstruction and include a table that shows the morphological differences between the current species and the two new species. For the Apiospora, have the authors compared the morphology of all the current species without DNA sequences to the two new species? If so, how are they different. This is a requirement and without a clear sentence stating this was completed, I cannot accept them as new species.
Please see attached for specific comments.

Author Response
Dear Editors and Reviewers:
Thank you for your letter and comments relating to our manuscript entitled “Diversity and new species of Ascomycota from bamboo in China” (ID: jof-3024840). The comments were very helpful in revising and improving our manuscript as well as emphasizing the significance to our research. We have read the comments carefully and made corrections accordingly. Revised portions are marked in blue in the manuscript. The main corrections in the paper and our responses to the reviewer’s comments are given below. We hope that the revisions in the manuscript and our accompanying responses will be sufficient to make our manuscript suitable for publication in the Journal of Fungi.
Responses to the comments of the reviewer:
Reviewer 3#
Comments 1: The manuscript is well prepared in most places but there are two places that need some grammar clarification (see attachment). The two major issues are 1) the authors reference Bifusisporella sichuanensis but it is not on the tree, and it appears that index fungorium has more Apiospora species then the authors mention that are on their tree. Since there are only a few Bifusisporella species it would be best to have the B. sichuanensis on the phylogenetic reconstruction and include a table that shows the morphological differences between the current species and the two new species. For the Apiospora, have the authors compared the morphology of all the current species without DNA sequences to the two new species? If so, how are they different. This is a requirement and without a clear sentence stating this was completed, I cannot accept them as new species.
Response 1: We have rechecked the errors throughout the whole manuscript and added spacing where it was missing. Bifusisporella sichuanensis is below the first red lettering on the tree, line 42 from top to bottom. In this article, we illustrate the differences between B. fujianensis and B. sichuanensis, B. bambooensis and B. sorghi in terms of DNA sequences, their molecular differences in terms of trees. Addtionally, a table of their differences in form is also added. Thank you for your wonderful comments!
Comments 2:Please see attached for specific comments.
Comments 2.1: Typically all words in the title are key words. I would recommend removing this one and adding another. Potentially the two genera.
Response 2.1: We have revised them.
Keywords: Apiospora; Bifusisporella; bambusicolous fungi; molecular phylogeny; morphology; new species
Comments 2.2: The being of this sentences needs to be looked over for grammar. It is missing something.
Response 2.2: We have revised them.
In this study, four fungal isolates from infected bamboo plants were identified and placed within the Magnaporthaceae, and their taxonomic relationships were clarified based on morphological characteristics and molecular identification; the latter involved multilocus phylogenetic reconstructions using a combined dataset of the intervening 5.8S nrRNA gene (ITS), the 28S large subunit of nuclear ribosomal RNA gene (LSU), the large subunit of RNA polymerase I (rpb1), the translation elongation factor 1-α gene (tef1-α) nucleotide sequences.
Comments 2.3: adjust the degree sign to match others.
Response 2.3: We have revised them.
Comments 2.4: Be consistent. In the first sentence of the paragraph this is spelled out, and it has already been abbreviated in the previous paragraph.
Response 2.4: We have revised them.
Comments 2.5: Where is this species in the tree.
Response 2.5:
Comments 2.6: Also, the author's will need a morphological difference. Sequence difference alone is not sufficient for the taxonomic code.
Response 2.6: We have revised them.
Morphologically, asexualmorph of B. sichuanensis was not observed and sexualmorph of B. fujianensis was not observed.
Comments 2.7: Where is this on the tree?
Response 2.7: In Bifusisporella bambooensis note, Bifusisporella sichuanensis should be Bifusisporella bambooensis. We have revised them.
Comments 2.8: Will need a morphological difference from this one to describe it a new species.
Response 2.8: We have revised them.
Compared to Bifusisporella sorghi, Bifusisporella bambooensis sp. nov. has larger conidiogenous cells and conidia (7.2-21.0 × 4.2-6.4 vs. 5.0–19.5 × 3.0–4.0 μm;10.8-45.0 × 2.8-4.9 vs. 19.0–34.0 × 3.0–4.0 μm)…
Comments 2.9: A table showing the differences would be beneficial. need to state what the differences are.
Response 2.9: We have added Table 3, which contains main morphological characters of Bifusisporella.
Table 3 The location, hosts or substrate, and main morphological characters of Bifusisporella
|
Species |
Location |
Host/substrate |
Conidiogenous Cells |
Size of Conidiophore Cells (µm) |
Conidia |
Size of Conidia (µm) |
References |
|
Bifusisporella bambooensis sp. nov. |
China |
Bambusoideae sp. |
cylindrical |
7.2-21.0 × 4.2-6.4 |
falcate or curved moon-shaped |
10.8-45.0 × 2.8-4.9 |
In this study |
|
Bifusisporellasorghi |
Brazil |
Sorghum bicolor |
cylindrical orclavate |
5.0–19.5 × 3.0–4.0 |
falcate |
Macroconidia 19.0–34.0 × 3.0–4.0 Microconidia 7.0–14.5 × 1.0–2.0 |
[41] |
|
Bifusisporella fujianensis sp. nov. |
China |
Bambusoideae sp. |
Cylindricalor rod-shaped |
8.9-14.3 × 5.8-8.1 |
falcate or curved moon-shaped |
37.3-56.3 × 3.6-5.7 |
In this study |
|
Bifusisporella sichuanensis |
China |
Phyllostachys edulis |
- |
- |
- |
- |
[42] |
Comments 2.10: Are these all the species? The author's need to examine all species that do not have molecular data.
Response 2.10: We have modified the sentence.
We tried our best to improve the manuscript and made some changes marked in blue in revised paper which will not influence the content and framework of the paper. We appreciate for Editors/Reviewers’ warm work earnestly and hope the revision will meet with your approval. Once again, thank you very much for your comments and suggestions.
Kind regards,
Junzhi Qiu
E-mail address: junzhiqiu@126.com